# Beyond Individual Differences in Affective Symptomatology: The Distinct Contributions of Emotional Competence and Rumination in a Nationally Representative Sample

**DOI:** 10.3390/bs15030318

**Published:** 2025-03-06

**Authors:** Ruth Castillo-Gualda, Juan Ramos-Cejudo

**Affiliations:** 1Faculty of Health Sciences, HM Hospitals, University Camilo José Cela, 28692 Villafranca del Castillo, Spain; jramos@ucjc.edu; 2HM Hospitals Health Research Institute, 28015 Madrid, Spain

**Keywords:** emotional competence, rumination, emotional regulation, affective symptomatology, anxiety, depression, well-being

## Abstract

Prior evidence suggests mental health, and affective symptomatology in particular, are influenced by emotion-related abilities. The strategies people use to identify, understand, and manage their emotions can serve as a protective or vulnerability factor for their psychological adjustment. Adaptive emotion regulation strategies, such as the ability to identify and understand emotions, can mitigate anxiety and depression symptoms, whereas maladaptive strategies, such as rumination, contribute to the vulnerability to suffering emotional symptomatology. To better understand the role of each strategy in affective outcomes, this study examines the role of adaptive emotion regulation strategies on anxiety, depression, and well-being, in comparison to rumination, among the general population. We hypothesize that the ability to accurately identify emotions and understand inner thoughts will act as protective factors against depression and anxiety, considering neuroticism. In comparison, rumination reactions, such as passively overthinking about inner experiences, will exacerbate the affective symptomatology. A sample of 1703 participants over the country, 50.43% female (18–75 years of age, mean = 45.48, SD = 14.73), closely aligned with the mean age of the target population in Spain, which is 44.1 years, were randomly selected to participate in this cross-sectional study. Participants completed self-report measures for emotional competence, rumination, anxiety and depression symptoms, well-being, and neuroticism. Structural equation modeling (SEM) was used to explore the relationships among the above-mentioned variables. Our results revealed neuroticism is related to higher levels of anxiety and depression and negatively related to wellbeing through the mediation effect of rumination and emotional competence, including all possible paths of the mediation model. This study has important implications for designing preventive and therapeutical interventions.

## 1. Introduction

Mental health problems represent a pressing global public health concern due to their increasing prevalence and the significant burden they place on healthcare systems. At the international level, recent studies have shown that mental disorders affect up to 20% of the global adult population, with a sustained rise over the past few decades, particularly following the COVID-19 pandemic ([38]; [46]). This trend is especially alarming in OECD countries, where depression rates among young adults have increased exponentially in recent years ([28]; [27]). Notably, recent studies also highlight that younger populations are particularly vulnerable to these mental health challenges ([36]). Prior evidence supports mental health issues often emerge during adolescence and tend to evolve over time, manifesting as different symptoms and diagnoses throughout life ([4]). These longitudinal trends underscore the importance of addressing mental health problems from a developmental perspective. Given this rising prevalence of mental health issues, it is crucial to investigate the underlying psychological factors that contribute to individual differences in mental health outcomes. Recent evidence has provided support for the Hierarchical Taxonomy of Psychopathology (HiTOP) ([11]). This framework systematically examines the relationships among symptoms, personality traits, and maladaptive behaviors as central contributors to the development and maintenance of mental health disorders such as anxiety and depressive disorders.

### 1.1. The Role of Personality

There is well-established evidence demonstrating personality has a significant relationship with psychological adjustment. Neuroticism is characterized by a heightened tendency to experience negative emotions, such as anxiety, fear, irritability, and sadness, with greater frequency and intensity ([23]). This emotional predisposition has been consistently associated with an increased risk of developing mental disorders, particularly depression and anxiety ([29]). Previous evidence highlights neuroticism as a robust predictor of affective disorders. For example, research suggests that people with high levels of neuroticism are up to three times more likely to develop major depressive disorder during their lifetime ([18]). Furthermore, this personality trait significantly contributes to psychological vulnerability in stressful contexts, amplifying negative emotional responses and limiting adaptive coping capacities ([21]). In the domain of anxiety, neuroticism is strongly linked to heightened reactivity to perceived threats and a greater reliance on maladaptive coping strategies, which, in turn, perpetuate anxiety symptoms ([25]). Importantly, neuroticism not only predicts the onset of mental health problems but is also associated with the chronicity and resistance to treatment of these disorders ([20]). These findings underscore the importance of considering neuroticism in both clinical assessments and interventions aimed at mitigating psychological vulnerability and enhancing adaptive/maladaptive strategies. Taken together, the evidence positions neuroticism as a key vulnerability factor. However, beyond personality traits, it is reasonable to assume the presence of mediating variables that could either amplify or mitigate emotional symptoms.

### 1.2. Emotion Regulation Strategies and Affective Symptomatology

This trait might be influenced by a range of psychological and social factors that shape emotional balance. In this context, emotional regulation emerges as a key variable in the development of psychological disorders, particularly those involving affective symptomatology ([1]).

Emotional competence is a set of skills, knowledge, and attitudes related to the process of emotional information and these competences have been extensively associated with mental health outcomes as well ([39]). Emotional identification is the ability to recognize emotions in oneself and others, including in complex or ambiguous contexts. Emotional understanding involves interpreting the meaning of emotions by linking them to triggering circumstances or events and understanding how emotions influence behaviors. The existing literature has shown that the ability to identify and understand emotions is negatively associated with depressive symptoms (*r* = −0.16) and anxiety (*r* = −0.15), and these competencies are linked to a variety of healthy habits with moderate effect sizes ([24]). People with high emotional competence tend to manage stress more effectively and employ more adaptive strategies for coping with complex emotional situations, reducing their vulnerability to developing emotional disorders and improving life satisfaction. The data indicate that emotional skills are negatively associated with stress (*r* = −0.66) and positively associated with life satisfaction (*r* = 0.54), with moderate-to-large effect sizes ([37]). 

One of the most well-established models for understanding emotional regulation strategies is the process model ([12]). This model classifies a series of strategies, some focused on cognitive aspects and others on expressive ones, aimed at managing emotions. According to Gross, emotional regulation is defined as “the process by which people influence the emotions they experience, when they experience them, and how they are expressed”. Although multiple strategies may occur simultaneously, the explanatory value of this model lies in its ability to identify the effectiveness of each strategy across various components of an emotional experience, such as the cognitive, physiological, or expressive elements. The scientific literature has identified various adaptive and maladaptive strategies with differentiated effects on mental health ([14]; [33]; [41]). Among the most relevant are cognitive change strategies, which involve reinterpreting the value attributed to the trigger event. Specifically, cognitive reappraisal—the ability to analyze a triggering situation from an alternative perspective—is a key tool in emotional regulation. It is one of the most scientifically supported strategies due to its strong association with psychological adjustment and individual well-being ([1]), being linked to better emotional adjustment and lower levels of anxiety and depression ([15]). In general, adaptive emotional regulation strategies are negatively associated with depressive symptomatology. Aldao’s meta-analysis (2010) reported that cognitive reappraisal was negatively associated with symptoms of depression (*r* = −0.17) and anxiety (*r* = −0.13), with small-to-moderate effect sizes. More recently, a systematic review and meta-analysis found that less frequent use of cognitive reappraisal was associated with anxiety symptoms (*r* = −0.30 to −0.50) and depressive symptoms (*r* = −0.56 to −0.70), with moderate-to-large effect sizes (*d* = 0.28–1.19) ([9]; [34]).

Conversely, maladaptive strategies, such as rumination and avoidance, are strongly linked to a higher risk of developing emotional disorders. Rumination is a cognitive reaction to unpleasant emotional experiences characterized by sustained and repetitive focus on negative thoughts, often related to the triggering event or emotionally intense situations. This response prevents adequate emotional processing, leading to intensified negative emotional symptoms ([26]). Specifically, rumination has been identified as one of the most prominent risk factors for depression and anxiety, with moderate associations (*r* = 0.55) ([1]). Among maladaptive strategies, rumination is one of the most empirically supported responses related to mental health issues ([10]; [22]). Rumination not only prolongs depressive episodes but also increases their severity and the likelihood of relapse by hindering engagement in pleasurable activities or resolving the underlying issues causing distress. Additionally, rumination acts as a barrier to effective depression treatment, interfering with the efficacy of cognitive-behavioral therapies by complicating the adoption of new coping strategies ([44]). Longitudinal studies have shown that people with a greater tendency to ruminate are more likely to develop depressive and anxiety disorders in the future, due to their inability to redirect attention away from negative thoughts ([14]; [45]).

In summary, the evidence suggests that people who predominantly use adaptive emotional regulation strategies, such as the ability to identify, understand, and reframe emotionally intense situations, tend to experience greater emotional well-being. In contrast, reliance on maladaptive strategies, such as rumination, increases the risk of developing emotional disorders like depression and anxiety ([13]). High levels of rumination, compared to more adaptive strategies such as cognitive reappraisal, negatively impact someone’s daily emotions when facing stressful situations by capturing attentional control and eventually intensifying emotional response ([35]). Recent research has advanced the understanding of these associations, showing that negative cognitive bias could mediate the relationship between neuroticism and difficulties in negative emotion regulation, accounting for 90.42% of the total effect ([6]). This finding may elucidate why people with high levels of neuroticism are more prone to engage in overthinking or exhibit reduced self-distancing when faced with challenging situations. Prior research on protective and vulnerability factors contributing to mental health has shown personality traits and emotional regulation strategies are closely linked to affective symptomatology. However, few studies have comprehensively examined the specific contributions of both protective and vulnerability factors to these affective symptoms, as well as their relationship with self-reported well-being, in a nationally representative sample.

### 1.3. The Present Study

This study aims to analyze how neuroticism explains higher levels of anxiety, depression, and lower subjective well-being, through the mediation effect of emotional abilities and strategies. Therefore, maladaptive strategies may contribute to the development of heightened affective symptomatology, particularly depression and anxiety. Conversely, emotional competence—specifically the ability to identify and understand emotions—may contribute to reducing affective symptomatology and enhancing subjective well-being. We expect to find that (1) emotional competence will have a strong negative correlation with symptoms of depression and anxiety, (2) rumination will exhibit a strong positive correlation with these affective symptoms, and (3) structural equation modeling (SEM) will reveal the specific and individual contributions of adaptive (e.g., emotional competence) and maladaptive (e.g., rumination) emotional variables to symptoms of anxiety and depression, as well as subjective well-being. Both adaptive and maladaptive strategies will make unique contributions to these affective variables. This would suggest that understanding emotional balance and mental health requires considering the full spectrum of adaptive and maladaptive strategies people employ and exploring how these strategies might mitigate or exacerbate the influence of personality traits such as neuroticism (Figure 1).

## 2. Materials and Methods

### 2.1. Participants and Procedure

The sample in this cross-sectional study was made up of 1703 participants from the general population of Spain, including every single region all over the country. They ranged from 18 to 75 years of age (M = 45.48, SD = 14.73). Women made up 50.43% of the sample, and men 49.57%. This representative of the national population sample was recruited as a part of a study based on psychological factors involved in the country’s mental health by the company MDK market research, from the 4th to the 8th of September 2023. Sampling error: ±2.37%~3.1% for a disproportional sample with a 95% confidence level.

The questionnaires were administered electronically and completed individually, thus ensuring the anonymity of answers. People who participated in the study were rewarded with EUR 20 each. Participation was fully voluntary and anonymous. This study was approved by the ethics committee following ICC/ESOMAR for market studies and the Declaration of Helsinki ([47]) as a statement of ethical principles for medical research involving human participants.

### 2.2. Instruments

The *Patient Reported Outcomes Measurement Information System* (PROMIS; [5]) assesses, through different measures, the physical, mental, and social health state in adults and infants, as well as in the general population and people with chronic diseases. In this study, we used specifically the PROMIS depression domain (PROMIS Depression [PROMIS-D]) and the PROMIS anxiety domain (PROMIS anxiety [PROMIS-A]), shortened 8-item version and Spanish adaptation by [43] ([43]). PROMIS-D bank items allow us to assess the presence of a depressive emotional state among the participants over the seven days prior to the measurement. The items are in a Likert-type format, where 1 means “never” and 5 means “always”. Total scores can range from 8 to 40, with higher scores indicating a greater presence of depressive symptomology. Both the original PROMIS questionnaire and the Spanish adaptation, and both the original 28-item version and the shortened 8-item version, have been shown to have very good psychometric properties ([43]). PROMIS-A bank items allow us to assess the presence of anxiety symptoms over the seven days prior to the measurement. The items are in a Likert-type format, where 1 means “never” and 5 means “always”. Total scores can range from 8 to 40, with higher scores indicating a greater presence of anxiety symptomology as well. In this study, PROMIS-D and PROMIS-A have reached excellent internal consistency reliability (see Table 3).

*Big Five Inventory—44* ([2]). The 44-item BFI comes from the original version of the BFI ([17]) and uses short phrases to assess the most prototypical traits associated with each of the Big Five dimensions in English ([17]). The trait adjectives (e.g., thorough) that form the core of each of the 44 BFI items (e.g., “does a thorough job”) have been shown in previous studies to be univocal, prototypical markers of the Big Five dimensions ([16], [17]). Participants rate each BFI item on a 5-point scale ranging from 1 (disagree strongly) to 5 (agree strongly); scale scores are computed as the participant’s mean item response (i.e., adding all items scored on a scale and dividing by the number of items on the scale). The Spanish version has been shown to have very good psychometric properties ([2]). We selected only the neuroticism subscale (items 4, 6, 9, and 14). In this study, we find excellent internal consistency reliability (see Table 3).

*The Profile of Emotional Competence* (PEC; [3]). The full 50-item PEC comprises 10 subscales forming 2 factors (intra- and inter-personal of emotional competence) and 1 global emotional intelligence (EI) score. It was developed to measure intra-personal EI and inter-personal EI separately. It assessed the five core emotional competences (identification, understanding, expression, regulation, and use of emotions) distinctly for one’s own emotions and for the emotions of others. It has been validated in several studies on a total of nearly 22,000 subjects. We selected 6 items from the subscales of identification and understanding emotions to evaluate the way individuals identify their emotional states and the way they understand what they are feeling. In this study, we find excellent internal consistency reliability (see Table 3).

*Rumination—Reflection Questionnaire; Rumination Subscale* (RRQ; [42]). This questionnaire assesses, regardless of affective state, the tendency to use rumination and reflection about past events. In this study, we only used the rumination subscale (e.g., “I spend a great deal of time thinking back over my embarrassing or disappointing moments”), with 12 items. Items are rated on a 5-point Likert scale ranging between 1 = “Strongly disagree” and 5 = “Strongly agree”, and the total score ranges from 12 to 60. The original study showed good internal consistency for both scales. We used a Spanish version of the RRQ ([32]), which has shown adequate psychometric properties. In this study, we find excellent internal consistency reliability (see Table 3).

*The Satisfaction with Life Scale* (SWLS: [7]) is probably the most cited life satisfaction measure in the scientific literature ([8]). Until its creation, most existing scales of subjective well-being focused on the emotional component ([30]). The scale was generated from a set of 48 self-report items related to life satisfaction and positive and negative affect. A factor analysis revealed three factors (life satisfaction, negative affect, and positive affect). The satisfaction factor contained ten items that were eventually reduced to the five that compose the current scale based on a semantic similarity analysis ([7]). Thus, the SWLS is a short scale composed of five simple items. The good psychometric properties of the SWLS have been confirmed over the past twenty years ([31]), with high internal consistency of the scale and Cronbach’s alpha coefficients ranging from 0.79 to 0.87 ([7]). We used item number 3 for this study, “I am completely satisfied with my life”.

### 2.3. Statistical Analyses

We used the software program JASP 0.19.1.0. to analyze the descriptive statistics, to calculate reliability and Pearson correlation coefficients, and for the structural equation modeling (SEM). We used SEM analysis with observed and latent variables to analyze the specific association between adaptive (emotional competence) and maladaptive (rumination) emotion regulation strategies, on the one hand, and depressive symptoms, anxiety symptoms, and well-being on the other. The latent variable of well-being was created by using item 3 of the SWLS, and item 2 from the sociodemographic questionnaire: “have you ever asked for professional help in mental health”. Even recommendations for building latent variables point to using at least three items, following [49] ([49]) and [48] ([48]). Given correlations from −0.66 to 0.88 with each other, a factor with 2 variables was considered reliable. This type of analysis makes it possible to control measurement errors. We performed the best model fit for the data, taking this as a limitation.

In accordance with [40]’s ([40]) recommendations, the following measurements of goodness of fit were used: (a) root mean square error of approximation by degrees of freedom (RMSEA); (b) Bentler’s comparative fit index (CFI); and (c) standardized root means square residual (SRMR). For the CFI, values over 0.90 indicate an acceptable goodness of fit. RMSEA values under 0.09 are acceptable, while values under 0.05 indicate a good level of fit. Finally, SRMR values are expected to be under 0.10 ([40]). The sample size in this study was calculated to be large enough for the use of SEM analysis, according to the recommendation made by [19] ([19]).

## 3. Results

### 3.1. Descriptive Analyses, Correlations and Reliability

Table 1 and Table 2 show the sociodemographic characteristics of the sample. Reliability and correlations are shown in Table 3. All the scales used had an acceptable level of internal consistency, with Cronbach’s alpha scores ranging from 0.77 to 0.96, except for well-being, given the two-item solution with a Cronbach’s alpha of 0.35.

There was a positive association between neuroticism and anxiety and depressive symptoms, as well as a negative association with well-being. There was also a negative correlation between emotional competence and rumination as we expected. There was a positive association between emotional competence and well-being, as well as a negative association between emotional competence and anxiety and depressive symptomology, indicating that participants who more frequently used adaptive emotion regulation strategies experienced lower levels of rumination, anxiety, and depression. Meanwhile, maladaptive strategies such as rumination displayed a large and positive correlation with symptoms of depression and anxiety. On the other hand, age was negatively correlated with negative outcomes such as rumination, anxiety, and depression.

### 3.2. Structural Equation Modeling (SEM)

We tested the proposed model in which the neuroticism trait is related to higher levels of anxiety and depression and negatively related to well-being through the effect of rumination and emotional competence, including all possible paths of the model. The model showed the following fit indices. The SEM results indicated a good fit: χ^2^ = 16.365, df = 56, *p* < 0.01, RMSEA = 0.09; CFI = 0.99; SRMR = 0.017, as well as Mardia’s coefficient (*p* < 0.001). See Table 4 for regression coefficients.

As Figure 2 shows, neuroticism is related to higher levels of anxiety and depression and negatively with well-being, through the effect of rumination and emotional competence, including all possible paths of the model. Anxiety and depression were positively and strongly associated and negatively associated with well-being.

In sum, in the proposed model, the neuroticism trait is related to higher levels of affective symptomatology (anxiety and depression) and is negatively associated with well-being through the effects of rumination and emotional competence as emotion regulation strategies.

## 4. Discussion

In this study, we aimed to understand the mechanisms underlying the association between neuroticism and affective symptomatology using a nationally representative sample. We examined the relevance of neuroticism and emotion regulation strategies in clinical affective symptoms through a cross-sectional approach. We expected that neuroticism had a specific contribution to maladaptive emotion regulation strategies in clinical affective symptoms. Our results are in line with these predictions.

First, the results of the correlation analyses confirm our hypothesis, as they show that there was a negative association between adaptive emotional regulation strategies and anxiety and depressive symptoms, while the association between maladaptive strategies and these symptoms was strong and positive. There was a negative association between neuroticism and adaptive emotion regulation strategies, as well as a positive relationship between this trait and maladaptive emotion regulation strategies and affective symptomatology. These results echo some of the main findings in the reviewed literature ([1]). Looking into this deeper, we found a consistent pattern of results for the strategies assessed. The findings highlight the role of adaptive emotional strategies—specifically, the ability to identify and understand emotions—as a key protective factor for mental health. These strategies enable people to effectively navigate emotionally charged situations and recognize their affective responses. Similarly, people with higher emotional competence show a more objective and adaptive perspective when interpreting emotionally intense situations. In contrast, rumination emerges as a critical vulnerability factor, reinforcing negative emotional cognitive cycles and interfering with healthier coping skills in emotionally triggering situations. Second, the results of the SEM analyses indicated that adaptive and maladaptive strategies both made specific contributions to affective symptomatology and well-being. The mediating role of emotional competence and rumination in the association between neuroticism, affective symptomatology, and subjective well-being underscores the importance of targeting these factors for both preventive and clinical interventions. Addressing emotional competence as a protective factor and rumination as a vulnerability factor may significantly contribute to the promotion of mental health and overall well-being.

Taken as a whole, our study aligns with previous research in the field ([14]; [33]; [41]). A substantial body of evidence suggests that people with high emotional competence are more likely to use adaptive emotion regulation strategies, exhibit enhanced identification of emotional signals, and demonstrate a more accurate interpretation of emotional triggers—all of which have significant implications for their mental health ([1]). People who tend to use these adaptive emotion regulation strategies do so because they are more competent emotionally ([37]; [39]). This study supports the idea that the ability to accurately identify the information conveyed by emotions, along with the use of cognitive reappraisal strategies to effectively reinterpret and reframe challenging life situations, helps increase subjective well-being and reduce the risk of developing affective symptomatology. In contrast, the tendency to dwell on distressing thoughts and emotions, and the reduced ability to shift attention toward more adaptive perspective taking, leads to the intensification of unpleasant moods.

These results have implications for both preventive and clinical settings. In relation to preventive strategies, mental health issues frequently emerge during adolescence and tend to evolve over time, manifesting as diverse symptoms and diagnoses across the lifespan ([4]). This highlights the critical importance of implementing interventions aimed at educating adaptive emotional skills early in life. Developing protective abilities, such as emotional competence (understood as the capacity to identify, understand, and manage emotions and thoughts) can buffer against the adverse effects of certain personality tendencies ([29]). By fostering these skills during adolescence, it may be possible to reduce the risk of ineffective emotional strategies and mitigate the progression of affective symptomatology, such as anxiety and depression, throughout development. Regarding clinical settings, both pharmacological and psychological treatments are especially effective at treating affective symptoms, but they have faced difficulties when it comes to reducing relapses. Our findings suggest that treatments aimed at reducing the use of maladaptive strategies (such as rumination) and encouraging more adaptive strategies (such as identifying and understanding emotions) might help improve the overall affective state of people with high levels of neuroticism given a disorder came up, reducing the risk of therapeutic failure ([44]).

Our study has some limitations. Firstly, although the methodology used made it possible to assess the effects of each of the variables and to reduce measurement error, this remains a cross-sectional study. This means that it is not possible to draw conclusions as to causality regarding the associations we found here. Therefore, it is necessary to attempt to replicate these results using experimental or prospective study designs, which would bring us closer to establishing causality among these variables. There are remarkable implications for the low internal consistency for the well-being latent factor, and we might consider improving this variable’s measurement in future studies. Secondly, the use of a community sample means that the results cannot be generalized to a clinical population. Thus, it would be desirable to explore whether similar results would be found among participants showing clinical disorders.

In conclusion, this study shows the specific contributions of adaptive and maladaptive strategies to affective symptomatology. Moreover, people who become competent to identify, understand, and manage their emotions can mitigate the impact of certain personality traits, such as neuroticism. Specifically, greater awareness of emotions such as stress or sadness as they arise, and a better understanding of emotional signals and situations, allow them to better evaluate the influence of these emotions on their decision-making and reasoning processes. Encouragingly, all these emotional skills and strategies can be improved and developed in educational settings (i.e., school), starting in childhood and at any stage of life.

## Figures and Tables

**Figure 1 behavsci-15-00318-f001:**
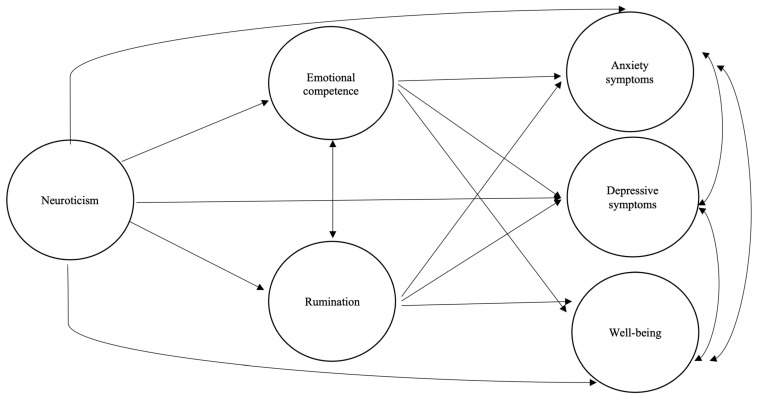
An exploratory testing hypothesis of specific associations between traits, emotion regulation strategies, and affective symptoms.

**Figure 2 behavsci-15-00318-f002:**
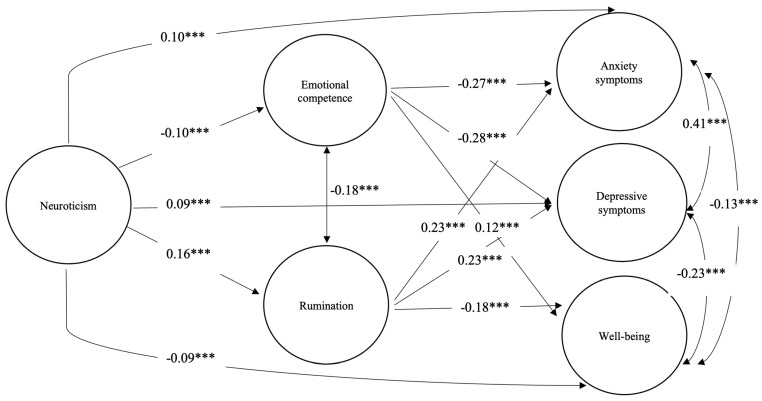
Structural equation model (SEM) of specific associations between traits, emotion regulation strategies, and affective symptoms. Note: Standardized beta coefficients are shown. *** *p* < 0.001.

**Table 1 behavsci-15-00318-t001:** Sociodemographic characteristics of the sample.

	N	%
Gender		
Male	843	49.57%
Female	857	49.57%
Age		
18–29	285	16.79%
30–44	491	28.90%
45–64	676	39.77%
65–75	247	14.54%
Education		
No studies	12	0.72%
Basic Studies	203	11.91%
High School	652	38.55%
University	833	49.01%
Employment Status		
Employed	1124	66.13%
Unemployed	173	10.20%
Studying	102	5.99%
Retired	233	13.72%
No work, no study	67	3.95%
Annual Income		
<1000/m	199	11.70%
1001–2000/m	525	30.88%
2001–3000/m	438	25.79%
3001–4000/m	224	13.19%
4001–5000/m	114	6.69%
More than 5000	98	5.79%
Unknown	101	5.94%

**Table 2 behavsci-15-00318-t002:** Sample distribution by region and gender.

	Male	Female	n	%
Andalucia	152	154	306	18.00%
Aragon	24	24	47	2.79%
Asturias	18	19	37	2.20%
Balearic Islands	23	22	45	2.65%
Canary Islands	42	43	85	4.99%
Cantabria	10	11	21	1.24%
Castilla y Leon	43	42	86	5.03%
Cstilla la Mancha	37	36	73	4.28%
Catalonia	135	138	273	16.08%
C Valenciana	90	91	182	10.68%
Extremadura	19	19	38	2.25%
Galicia	48	49	97	5.72%
Madrid	117	126	244	14.33%
Murcia	27	27	54	3.15%
Navarra	12	12	23	1.37%
Basque Country	38	40	78	4.60%
Rioja	6	6	11	0.66%

**Table 3 behavsci-15-00318-t003:** Means, standard deviations, internal consistency reliabilities, and intercorrelations among psychological measures.

	1	2	3	4	5	6	7	8	9
1. Age	-								
2. Neuroticism	−0.18 ***	-							
3. Rumination	−0.13 ***	0.57 ***	-						
4. Identification	0.16 ***	−0.09	0.00	-					
5. Understanding	0.16 ***	−0.40 ***	−0.37 ***	0.19 ***	-				
6. Anxiety	−0.22 ***	0.60 ***	0.52 ***	−0.10 ***	−0.56 ***	-			
7. Depression	−0.20 ***	0.53 ***	0.48 ***	−0.11 ***	−0.53 ***	0.84 ***	-		
8. Well-being	0.145 *	−0.50 ***	−0.42 ***	0.19 ***	0.27 ***	−0.50 ***	−0.56 ***	-	
9. Emotional competence	0.208 ***	−0.35 ***	−0.28 ***	0.67 ***	0.85 ***	−0.47 ***	−0.46 ***	0.30 ***	-
MD	45.48	10.18	40.17	15.66	11.68	21.99	20.38	8.25	27.9
DT	14.73	2.88	7.51	3.47	4.93	9.72	10.21	1.75	6.55
α	NC	0.79	0.77	0.84	0.87	0.95	0.96	0.35	0.77

* *p* < 0.05, *** *p* < 0.001.

**Table 4 behavsci-15-00318-t004:** SEM parameter estimates.

						95% Confidence Interval		Standardized		
Outcome	Predictor	Estimate	Std. Error	z-Value	*p*	Lower	Upper	All	LV	Endo
Rumination	Neuroticism	1.292	0.044	29.080	<0.001	1.205	1.379	0.576	1.292	0.164
Anxiety	Neuroticism	1.001	0.062	16.033	<0.001	0.878	1.123	0.364	1.001	0.103
	Rumination	0.290	0.027	10.909	<0.001	0.238	0.342	0.237	0.290	0.237
	Emotional comp.	−0.413	0.028	−14.754	<0.001	−0.468	−0.358	−0.280	−0.413	−0.280
Depression	Neuroticism	0.875	0.069	12.628	<0.001	0.739	1.011	0.303	0.875	0.086
	Rumination	0.300	0.030	10.172	<0.001	0.242	0.358	0.233	0.300	0.233
	Emotional comp.	−0.447	0.031	−14.387	<0.001	−0.508	−0.387	−0.289	−0.447	−0.289
Well-being	Neuroticism	−0.174	0.013	−13.338	<0.001	−0.199	−0.148	−0.350	−0.174	−0.099
	Rumination	−0.041	0.006	−7.353	<0.001	−0.052	−0.030	−0.184	−0.041	−0.184
	Emotional comp.	0.033	0.006	5.689	<0.001	0.022	0.045	0.125	0.033	0.125
Emotional comp.	Neuroticism	−0.661	0.042	−15.668	<0.001	−0.743	−0.578	−0.355	−0.661	−0.101

## Data Availability

Restrictions apply to the availability of these data. Data was obtained from Linea Directa and is available from the authors with the permission of Linea Directa.

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
