# Peer review of "Beyond Individual Differences in Affective Symptomatology: The Distinct Contributions of Emotional Competence and Rumination in a Nationally Representative Sample"

_behavsci, 2025, doi:10.3390/bs15030318_

Round 1
Reviewer 1 Report
Comments and Suggestions for Authors
Thank you for the opportunity to analyze and review this work. I would like to express my gratitude to the authors and the editor for their dedication in producing such a thorough and meaningful study. This work is a significant contribution to understanding how emotion regulation strategies influence mental health, and I particularly appreciate the focus on both preventive and clinical practical implications. Additionally, the choice of a representative sample and the use of structural equation modeling (SEM) to examine the relationships between emotional variables and affective symptomatology add great robustness to the study.
Strengths of the Work:
The literature review is comprehensive (lines 34-153), providing an exhaustive and useful theoretical framework. Furthermore, the methodology (lines 171-263) is detailed, demonstrating particular attention to the selection of tools and statistical analyses. Finally, the discussion section (lines 326-407) effectively highlights practical implications and offers recommendations for future interventions.
Suggested Modifications:
- Abstract (lines 8-32):
- Line 23: Specify the significance of the mean age (e.g., representativeness of the sample compared to the general population).
- Line 28: Include a sentence explicitly mentioning the study’s limitations (e.g., the cross-sectional nature of the research).
- Introduction (lines 34-153):
- Line 44: Provide more detail on the diagnostic criteria for the affective disorders mentioned to clarify the context.
- Line 75: Add a practical example to better illustrate the differences between adaptive and maladaptive strategies. For example, explain how rumination negatively impacts someone dealing with emotional stress compared to more functional strategies such as cognitive reappraisal.
- Materials and Methods (lines 171-263):
- Line 203: Offer a clearer explanation of the selection of specific items from the BFI scale (why these items specifically?).
- Line 221: Explain why certain items from the PEC were selected and how this aligns with the study’s objectives.
- Line 269: Address the low internal consistency for the measure of well-being (α = 0.35) and discuss its potential implications for the results.
- Results (lines 264-315):
- Line 313: Elaborate on the limitations of the SEM model, such as potential measurement biases or challenges in interpreting latent variables.
- Discussion (lines 326-407):
- Line 390: Emphasize the cross-sectional nature of the study as a limitation, highlighting the need for longitudinal studies.
- Line 401: Expand on the importance of developing adaptive emotional strategies in educational settings.
Sections to Consider Removing:
- Line 176: Mentioning the monetary reward for participants seems redundant, considering that anonymity and voluntary participation are already stated.
- Line 244: Reduce repetitive details about the tools, already explained in the preceding lines.
Analysis and Limitations:
The statistical analysis, supported by goodness-of-fit indices (lines 305-315), is robust. However, it would be beneficial to include further explanations about the implications of the low internal consistency for well-being and consider how to improve this variable’s measurement in future studies. Additionally, the limitations of the study are not sufficiently explored. The authors should emphasize how the cross-sectional nature limits causal inference and suggest longitudinal or experimental studies to confirm the findings.
Suggested Citations:
We kindly ask the authors to include the following articles:
- Petruccelli et al. (2014): This study should be cited in the discussion (lines 352-354), where the affective dimension and its implications for emotion regulation strategies are discussed. It could support the idea that emotion regulation is crucial for preventing emotional disorders.
- Full Citation: Petruccelli, F., Diotaiuti, P., Verrastro, V., Petruccelli, I., Federico, R., Martinotti, G., Fossati, A., Di Giannantonio, M., & Janiri, L. (2014). Affective dependence and aggression: an exploratory study. BioMed Research International, 2014, 805469. https://doi.org/10.1155/2014/805469
- Diotaiuti et al. (2021): This should be cited in the methodology section (line 215), concerning the validity of the psychometric tools used in the study.
- Full Citation: Diotaiuti, P., Valente, G., & Mancone, S. (2021). Validation study of the Italian version of Temporal Focus Scale: psychometric properties and convergent validity. BMC Psychology, 9(1), 19. https://doi.org/10.1186/s40359-020-00510-5
Reviewer 2 Report
Comments and Suggestions for Authors
1. It there are some studies that did the same how do you justify your study? " However, few studies have comprehensively examined the specific 150 contributions of both protective and vulnerability factors to these affective symptoms, as 151 well as their relationship with self-reported well-being, in a nationally representative sam- 152 ple. "
2. Please explain about the sample. Who is the author, who collected the data, how did they assured representativity and how they stratified the collection of the responses in order to assure representativity.
3. " except for wellbeing 268 given the two-item-solution with a Cronbach’s alpha of .35." if you had a measure with two items alpha Crombach shouldn't be performed. It needs at least 3 items.
4. Tables 4 and 5 can be written in a paragraph. There is no need to provide the tables form the software outputs.
5. SEM results are better to be presented in ta table with all coeficients.
6. The SEM Graph can be improved to look better and have only straight lines.
7. You can provide the empty SEM Graph in "The Present Study" section to create anticipation and show clearer what you want to test.
8. As I see you have no mediation here tested: "We tested the proposed model in which neuroticism trait is related with higher levels 306 of anxiety, depression and negatively with wellbeing through the mediation effect of ru- 307 mination and emotional competence, including all possible paths of the mediation model." If you wanna have mediations you must add some effects between neuroticism and anxiety symptoms and so on...
9. After the SEM Graph you have only the title of the graph. After should be some text. The transition is to abrupt.
10. For this statement and paragraph I am skeptical. "These results have important implications for both preventive and clinical settings. 372 ". This is just one study. More precaution is needed in speaking about implications.
11. "evidence" is a strong word. Line 400. "proven" line 383. These are strong words. Use indicate, shows, incline, ...
12. Years in bibliography must be written with bold.
Comments on the Quality of English Language
Adjustments needed:
Correction of strong words: prove, evidence
Recheck English.
Round 2
Reviewer 1 Report
Comments and Suggestions for Authors
We are therefore pleased to inform you that the manuscript, in its final version, is now ready to be accepted and published. We believe that this article represents a significant contribution to the field and will be of great interest to the readers.
Thank you for your commitment and collaboration during this revision process. Should there be any further steps required before official publication, we remain at your disposal for any final clarifications or adjustments.
Author Response
Dear Reviewer 1,
We sincerely appreciate your valuable and constructive comments. Following your suggestions, we believe our manuscript has significantly improved.
Thank you once again for your time and insightful feedback.
Best regards,
Reviewer 2 Report
Comments and Suggestions for Authors
It seems that some of the points were not addressed.
5. SEM results are better to be presented in ta table with all coeficients. Please present p values, unstandardized coeficients.... There are a few data about SEM analysis.
8. As I see you have no mediation here tested: "We tested the proposed model in which neuroticism trait is related with higher levels 306 of anxiety, depression and negatively with wellbeing through the mediation effect of ru- 307 mination and emotional competence, including all possible paths of the mediation model." If you wanna have mediations you must add some effects between neuroticism and anxiety symptoms and so on... You mention mediation model, buy there is not testing for such a test.
12. Years in bibliography must be written with bold and corrected according to the MDPI style.
Author Response
Dear Reviewer 2,
We sincerely appreciate your valuable and constructive comments. Following your suggestions, we believe our manuscript has significantly improved.
Please see the attachment to find the answers for your suggestions.
Best regards,
